# Effect of Acupuncture on Recovery of Consciousness in Patients with Acute Traumatic Brain Injury: A Multi-Institutional Cohort Study

**DOI:** 10.3390/healthcare11162267

**Published:** 2023-08-11

**Authors:** Chun-Chieh Lin, Hsing-Yu Chen, Chu-Yao Tseng, Chien-Chung Yang

**Affiliations:** 1Division of Acupuncture and Traumatology, Center for Traditional Chinese Medicine, Chang Gung Memorial Hospital, Taoyuan 33378, Taiwan; lin1122@cgmh.org.tw (C.-C.L.); boyau1125@cgmh.org.tw (C.-Y.T.); 2Graduate Institute of Clinical Medical Sciences, College of Medicine, Chang Gung University, Taoyuan 330036, Taiwan; 8705016@cgmh.org.tw; 3Division of Chinese Internal Medicine, Center for Traditional Chinese Medicine, Chang Gung Memorial Hospital, Taoyuan 33378, Taiwan; 4School of Traditional Chinese Medicine, College of Medicine, Chang Gung University, Taoyuan 33302, Taiwan

**Keywords:** acupuncture, traumatic brain injury, consciousness, Glasgow Coma Scale, cohort study, Chang Gung Research Database

## Abstract

Traumatic brain injury (TBI) causes cognitive dysfunction and long-term impairments. This study aims to examine the effectiveness of acupuncture on the recovery of consciousness in TBI patients. This is a retrospective, multi-institutional cohort study. We enrolled patients with newly diagnosed TBI from 1 January 2007 to 3 August 2021, aged 20 years and older, from the Chang Gung Research Database (CGRD). The outcome was defined based on the difference between the first and last Glasgow Coma Scale (GCS). A total of 2163 TBI patients were analyzed, and 237 (11%) received acupuncture in the treatment period. Generally, the initial GCS was lower in the acupuncture users (11 vs. 14). For the results of our study, a higher proportion of acupuncture patients achieved significant improvement (GCS differences ≥ 3) compared to non-acupuncture users (46.0% vs. 22.4%, *p*-value < 0.001). The acupuncture users had a 2.11 times higher chance of achieving a significant improvement when considering all assessable covariates (adjusted odds ratio (aOR) 2, 11, 95% confidence interval [CI]: 1.31–3.40; *p*-value = 0.002). Using 1:1 propensity score matching (PSM), the acupuncture users still had better outcomes than the non-acupuncture users (45.3% vs. 32.9%, *p*-value = 0.020). In conclusion, this study suggests that acupuncture treatment may be beneficial for TBI patients.

## 1. Introduction

Traumatic brain injury (TBI) generally refers to injuries such as intracranial hemorrhage or skull fracture caused by an indirect or direct physical or mechanical external force impact, which, in turn, causes brain dysfunction [1]. About 1.7 million people suffer from TBI, and at least 2.4 million emergency department visits, hospitalizations, and deaths are related to TBIs each year [2]. The mortality rate of TBI patients has significantly decreased due to the aggressive acute management and healthcare suggested in the past decades [3,4]. However, various residual disabilities such as impairments in cognitive, physical, behavioral, emotional, and social function issues may still happen subsequently [5,6]. Consciousness disturbance is the major complication after suffering from TBI [7]. TBI patients who have a consciousness disturbance have a greater residual disability, which could cause a huge burden on the healthcare system [7,8,9].

There are several tools or scales to grade the severity of the consciousness impairment in TBI, including the Glasgow Coma Scale (GCS). The GCS, which was proposed in 1974, is composed of three sub-categories: eye opening (E), verbal response (V), and motor response (M), with full marks of 4, 5, and 6 [10]. The total score is the sum of the three sub-categories, which can be used to reflect the disease severity, with 3–8 being severe, 9–12 being moderate, and 13–15 being mild [11,12]. Due to its convenience and intuition, it is still the most credible and widely used scale for clinically evaluating the severity of TBI. It may guide clinicians and investigators in optimizing their treatment decisions and predicting outcomes for patients with TBI.

Recent studies have suggested that neurorehabilitation and neurorestoration starting immediately after TBI play a vital role in consciousness recovery [13,14]. The conventional treatments range from medical management alone to radical surgical interventions based on the initial severity of the TBI. However, most of them are life-sustaining and prophylactic approaches that prevent disease deterioration and complications. Therefore, research on TBI should be shifted to neurorestoration [14]. There are limited interventions available in terms of promoting consciousness recovery [15]. Acupuncture is a major clinical treatment approach in traditional Chinese medicine (TCM). It has been widely used clinically in Asia to treat various diseases and has gradually expanded to other countries as a complementary and alternative medicine, with its use becoming more common in recent decades [16,17,18]. Acupuncture has been used as a feasible approach to consciousness recovery among TBI and stroke patients [19,20,21,22]. So far, only a small number of studies have been published on consciousness recovery with acupuncture treatment for TBI. In meta-analyses, Wong et al. and Tan et al. reported that acupuncture could improve GCS, motor and speech functions, and lower mortality after a TBI [19,20]. However, there is only limited evidence to support that acupuncture is effective for TBI. In addition, most of the designs of previous studies were less well-defined in the acute or chronic phase of TBI, leading to the effectiveness of acupuncture in the acute phase of TBI being undetermined.

This study aims to examine the effectiveness of acupuncture on the recovery of consciousness among TBI patients in the acute phase. The results of this study would clarify the feasibility of using acupuncture among acute TBI patients, especially when a disturbance in consciousness is evident.

## 2. Materials and Methods

### 2.1. Data Sources

This is a retrospective, multi-institutional cohort study. We used the Chang Gung Research Database (CGRD) to collect patients’ past medical records in the North Branch (Taipei, Linkou, Taoyuan) of the Chang Gung Memorial Hospital (CGMH) in Taiwan. The CGMH is the largest medical system in Taiwan, providing a large volume of medical services. Studies based on the CGRD are reported to be of a high quality and have been published in a diverse range of medical journals [23,24,25]. This database contains electronic daily medical records from the CGMH since 2001 and is an excellent source for conducting cohort studies, including personal data, a diagnosis of each admission, outpatient, or emergency visit, and co-morbidities. Records of daily treatments, managements, examinations, and consultation data sheets are also digitalized and stored in the CGRD prospectively.

### 2.2. Study Design

The study flow chart is demonstrated in Figure 1. The eligible subjects of our study had to meet the following inclusion criteria: 1. Aged 20 years and older. 2. Patients with newly diagnosed TBI who were hospitalized between 1 January 2007 and 3 August 2021, defined as inpatients with TBI as their main diagnosis and at least one of the following diagnosis codes as a first diagnosis during admission: ICD-9-CM: 800–804 and 850–854 and ICD-10-CM: S020–S021, S028–S029, S06, and S098–S099 [26,27,28,29,30]. Furthermore, TBI was classified as follows: skull fracture (ICD-9-CM: 800–804 and ICD-10-CM: S02.0–S02.1 and S02.8–S02.9), brain contusion (ICD-9-CM: 851 and ICD-10-CM: S06.30–S06.33), subarachnoid hemorrhage (SAH) (ICD-9-CM: 852.0–852.1 and ICD-10-CM: S06.6), subdural hemorrhage (SDH) (ICD-9-CM: 852.2–852.3 and ICD-10-CM: S06.5), epidural hemorrhage (EDH) (ICD-9-CM: 852.4–852.5 and ICD-10-CM: S06.4), intracranial hemorrhage (ICH) (ICD-9-CM: 853 and ICD-10-CM: S06.34–S06.38), and other (ICD-9-CM: 850 and 854 and ICD-10-CM: S06.0–S06.2, S06.89, S06.9, and S09.8–S09.9). The first day a subject was diagnosed with TBI during admission was set as the index date.

The severity of the consciousness impairment in the TBI was evaluated using the GCS. We identified the first GCS record of each subject within 7 days after the index date as the baseline for the initial disease severity, while the GCS on the 90th day after the index date was defined as the disease severity at the end of the follow-up in this study [31]. Subjects who died within 90 days after diagnosis, those with incomplete first and last GCS records, or GCS records that could not be quantified in terms of score (such as endotracheal intubation, tracheostomy, or aphasia, recorded as E1-4VeM1-6, E1-4VtM1-6, or E1-4VaM1-6) were all excluded.

Moreover, all the subjects were divided into acupuncture and non-acupuncture users. The subjects had to complete six sessions of acupuncture treatment to be defined as acupuncture users, whose first acupuncture record had to be after the first GCS date and not exceed the last GCS date. Subjects who did not receive acupuncture were defined as non-acupuncture users. Those who received less than six sessions of acupuncture treatment were screened out of the study. All the subjects received standard management after their diagnosis of TBI. The study protocol was approved by the Institutional Review Board of the Chang Gung Medical Foundation (No.: 202101351B0).

### 2.3. Covariates

Gender, age, initial severity, TBI type, hypertension, diabetes mellitus, hyperlipidemia, coronary artery disease (CAD), stroke, mental disease, dementia, and Parkinsonism were obtained as covariates in this study [3,32,33,34,35]. Age was reclassified into two categories: <50 and ≥50 years old. The severity of the baseline consciousness impairment was divided into severe, moderate, and mild groups according to GCS 3–8, 9–12, and 13–15, respectively. The co-morbidities of each subject were recognized by inpatient or outpatient visits within two years before the index date, with the following diagnostic codes: hypertension (ICD-9-CM: 401–405 and ICD-10-CM: I10–I15), diabetes mellitus (ICD-9-CM: 250 and ICD-10-CM: E08–E13), hyperlipidemia (ICD-9-CM: 272.0–272.4 and ICD-10-CM: E780–E785), CAD (ICD-9-CM: 410–414 and ICD-10-CM: I21–I25), stroke (ICD-9-CM: 433–438 and ICD-10-CM: I63–I67), mental disease (ICD-9-CM: 291–293 and 295–319 and ICD-10-CM: F06, F10–F19, F20, F22, F23, F25, F28, F29, F31–F33, F40, F41, F43, F51, and G47), dementia (ICD-9-CM: 290, 294, and 331 and ICD-10-CM: F01–F03, G30, and G31), and Parkinsonism (ICD-9-CM: 332 and ICD-10-CM: G20 and G21) [35].

### 2.4. Outcome

Our study used the difference between the first and last GCS as the primary study endpoint. To simplify the calculation, we reclassified the differences in the GCS into “Significant improvement” (GCS difference ≥ 3), “Stable condition” (GCS difference 0–2), and “Disease progression” (GCS difference < 0) [21,36].

For the secondary outcome, the difference in the motor scale in the GCS between the index and 90th day was assessed. For motor response improvement, a difference in motor score of ≥2 was classified as “Significant improvement”, a score difference of 0–1 was classified as “Stable condition”, and a score difference of <0 was classified as “Disease progression”.

### 2.5. Statistical Analysis

For the baseline characteristics, the χ^2^ test was used to analyze the categorical data. The numerical data are presented as medians and interquartile ranges (M(IQR)), and the differences between the acupuncture users and nonusers were examined using the Mann–Whitney U test.

For the outcome measurement, the χ^2^ test was used to analyze the difference in the subjects’ distribution between non-acupuncture and acupuncture users based on the subgroups of “Significant improvement”, “Stable condition”, and “Disease progression”. The logistic regression model was applied to compare the chances of two groups of subjects who had significant improvements in their GCS. Subjects with “Significant improvement” and those without were set as the dependent variable, and acupuncture use alone, not alone, or with all confounding factors were set as the independent variables in the univariable and multivariable regression analyses, respectively. Data are expressed as odds ratios (OR) and adjusted odds ratios (aOR) with a 95% confidence interval (95% CI) for the univariable and multivariable logistic regressions. The larger the number, the higher the chance of a subject having a significant improvement in their GCS. Furthermore, we performed a 1:1 propensity score-matched pair method to analyze the non-acupuncture and acupuncture users to reduce the confounding bias. The covariates selected for the pairing model were based on the result of the baseline characteristics. A *p*-value of <0.050 (two-tailed) was defined as a statistically significant difference. All the statistical analyses were performed using the commercial software SAS V.9.1 (SAS Institute Inc., Cary, NC, USA) and SPSS Statistics 24.0 (IBM, New York, NY, USA).

## 3. Results

### 3.1. Baseline Characteristics of TBI Patients Receiving/Not Receiving Acupuncture

Of the 2163 subjects, 237 subjects (11%) received acupuncture during this treatment period. In Table 1, the baseline GCS total score was lower in the acupuncture users (11 vs. 14). The baseline severity was divided into three subgroups: mild (13–15), moderate (9–12), and severe (3–8). The acupuncture users had a higher proportion in the moderate subgroup (37.6% vs. 24.1%) and severe subgroup (21.9% vs. 9.5%), but a lower proportion in the mild subgroup (40.5% vs. 66.4%), compared to the control group. For the TBI type and co-morbidities, the acupuncture group had a higher proportion of SDH (46.4% vs. 37.6%), ICH (38.4% vs. 22.7%), hypertension (22.4% vs. 15.9%), diabetes mellitus (13.1% vs. 9.0%), CAD (28.7% vs. 22.0%), and stroke (28.7% vs. 22.0%), compared to the control group. Instead, there was a lower proportion of skull fracture (23.2% vs. 41.0%) in the acupuncture users (Table 1).

After performing the 1:1 propensity score-matching procedure, a total of 468 subjects with 234 acupuncture and non-acupuncture users in each group were included. There were no significant differences in gender, age, initial GCS, TBI type, and all co-morbidities between the non-acupuncture and acupuncture users (Table 2).

### 3.2. Acupuncture Significantly Improves the Disturbance of Consciousness in TBI Patients

For all subjects, the use of acupuncture was associated with a higher improvement in consciousness impairment. “Significant improvement”: 46.0% vs. 22.4%; “Stable condition”: 43.9% vs. 68.8%; and “Disease progression”: 10.1% vs. 8.8% (*p*-value < 0.001). For the 1:1 PSM subjects, “Significant improvement”: 45.3% vs. 32.9%; “Stable condition”: 44.4% vs. 56.0%; and “Disease progression”: 10.3% vs. 11.1% (*p*-value = 0.020) (Table 3).

We further performed a logistic regression model to evaluate the chance of a significant improvement in the GCS between the non-acupuncture users and acupuncture users. The results are shown in Table 4, which indicate that the acupuncture users had a higher chance of a significant improvement in the GCS compared to the non-acupuncture users, both in all subjects (aOR, 2.01; 95% CI, 1.39–2.90; *p*-value < 0.001) and in the 1:1 PSM subjects (aOR, 2.11; 95% CI, 1.31–3.40; *p*-value = 0.002).

Table 5 indicates that the acupuncture users had a higher chance of a significant improvement in the GCS based on gender, age, and initial severity compared to the non-acupuncture users. After adjusting for all the covariates in Table 1, except age and initial severity, the subjects aged over 50 years old (aOR, 1.38; 95% CI, 0.87–2.18; *p*-value = 0.171) and the severe group (aOR, 1.63; 95% CI, 0.93–2.87; *p*-value = 0.088) had no higher chance of a significant improvement in the GCS, with no statistically significant difference between the non-acupuncture and acupuncture group.

### 3.3. Acupuncture Significantly Enhances the Motor Function Response in TBI Patients

For the motor response in all subjects, the use of acupuncture was associated with a higher improvement in motor response impairment. “Significant improvement”: 12.7% vs. 2.8%; “Stable condition”: 78.1% vs. 89.9%; and “Disease progression”: 9.3% vs. 7.3% (*p*-value < 0.001). For the 1:1 PSM subjects, “Significant improvement”: 12.4% vs. 3.0%.; “Stable condition”: 78.2% vs. 83.3%; and “Disease progression”: 9.4% vs. 13.2% (*p*-value < 0.001) (Table 6).

## 4. Discussion

To the best of our knowledge, this study is one of the few studies that has investigated the effectiveness of acupuncture in the acute phase of TBI, which is characterized by a rapid disease progression and high mortality. Our findings suggest that acupuncture treatment may be effective for consciousness recovery, and a combination with acupuncture treatment tends to reach better outcomes compared to conventional treatment alone. Furthermore, acupuncture treatment may be also effective for motor response recovery. Based on our findings, acupuncture may be regarded as an adjuvant therapy combined with conventional treatment through interprofessional collaborative practice with TBI specialists. Thus, we can put our findings into clinical practice considering this integrated care model. Additionally, acupuncture has the advantages of easy operating, rapid administration, and few adverse effects. More patients suitable for acupuncture treatment can complete evaluation and treatment promptly, thereby reducing healthcare costs while achieving a restoration of consciousness. In this study, we also adopted differences in motor responses as an indicator for improving the accuracy of level of consciousness grading, in addition to differences in the GCS, since motor responses are more sensitive than the GCS in detecting changes in the level of consciousness [37,38]. The present study can gain a more complete and comprehensive view while grading the level of consciousness. In addition, these findings can be more accurately and efficiently utilized to arrange treatment goals and evaluate outcomes in TBI patients.

We considered that a GCS score improvement of more than three was the definition of a significant improvement in clinical manifestations. This was according to our clinical experience and judgment. In addition, the results of previous studies that used the GCS as an indicator of effectiveness have demonstrated that a GCS improvement of ≥3 indicates a statistical significance compared to a control group. Therefore, we modified the existing grading method and reclassified the patients by the “difference of GCS” into three groups.

In Table 1, the acupuncture users had a higher proportion of subjects with a poor initial disease severity, SDH, ICH, hypertension, diabetes mellitus, CAD, and stroke. Previous studies have demonstrated that a poor GCS at presentation and co-morbidities are associated with poor disease outcomes [34,35], which also indicate that this group of subjects require more medical assistance and integrated care management. Based on this, our result revealed that acupuncture has become a feasible approach to treating consciousness disturbance under the conventional medical system. As for skull fracture, the present study showed a lower percentage of subjects in the acupuncture users compared to the non-acupuncture users. Our possible explanation for this is that multiple site fractures may often occur with skull fractures. Considering the feasibility of treatment, acupuncture may not be suitable for these patients, thus leading to a lower usage rate.

Our data were consistent with the results of the systemic reviews and meta-analyses conducted by Wong et al. and Tan et al. [19,20], which demonstrated that acupuncture could improve GCS score. However, neither of those studies mentioned the amount of GCS recovery. Our findings demonstrated that nearly 50% of the acupuncture users had more than three points of GCS improvement, suggesting that the amount of GCS recovery was significant. This finding was also consistent with the results of several RCTs that investigated the effect of acupuncture in the acute phase of stroke through changes in the GCS before and after treatment, which also revealed that an improvement in the GCS by at least three points was significant in the treatment group. The result from the RCT by Cai et al. using cross electro-nape acupuncture (CENA) treatments showed significant GCS improvements, with a mean of four points after 4 weeks of intervention in the GCS score in the treatment group [21]. However, they did not mention the co-morbidities of each group of subjects, which might have affected the outcome of the treatment. There is also limited evidence to support that the effect of CENA is superior compared to standard acupuncture. The RCT conducted by Li et al. also showed significant improvements in the GCS by a median of five points after 12 weeks of intervention in the acupuncture group [22]. Our baseline data showed that there was a lower percentage of female and higher CAD subjects compared to theirs. Previous studies have demonstrated that endogenous estrogen benefits the enhancement of the therapeutic influence of acupuncture on females, leading to better treatment outcomes [27]. Our subgroup analysis also had similar findings, with women having a higher chance of GCS improvement compared to men. In addition, CAD is associated with poor disease outcomes of TBI, leading to poor therapeutic effects [35]. These might be the possible explanations for their result being better than ours. Based on the above findings and interpretations, patients achieving a GCS recovery score of three can be used as an indicator of effectiveness, which might be used as a clinical guideline in treating TBI in the future. However, further research and clinical trials are still required to confirm our findings.

A cohort study conducted by Kowalski et al. demonstrated that motor response score is crucial and directly related to the disease severity of TBI [39]. Based on the results in Table 6 and Table 3, it is suggested that motor response was related to disease severity, further confirming the theory above. With the same classifications as our primary outcome, the result suggested that acupuncture could improve the motor response score in the acute phase of TBI, which was consistent with the findings of the RCT on acute-phase ICH subjects by Li et al. in [22], suggesting that acupuncture could improve motor response. Therefore, we could further hypothesize that the effect of consciousness recovery is mainly reflected in motor response recovery based on the above findings.

The ascending reticular activating system (ARAS) has been considered to be a main neural structure for consciousness, which is a pathway connecting the brainstem to the thalamus, the hypothalamus, and the basal forebrain [40]. As for the effectiveness of acupuncture, a previous study conducted by Park et al. demonstrated that acupuncture plays an important role in mediating brain neural activation and its functional connectivity, especially in the hypothalamus. The therapeutic effect of acupuncture on consciousness recovery may be mediated through ARAS activation [41]. For motor response recovery, evidence has suggested that there is a specific modulatory effect of acupuncture on the motor-related network of stroke patients [42]. Furthermore, a growing body of evidence indicates that acupuncture has beneficial effects on brain insults via many mechanisms, including anti-neuroinflammation, neuron repairment and regeneration, anti-apoptosis, angiogenesis, and reduced brain edema [43,44,45,46,47]. These findings could support the effect of acupuncture on consciousness recovery in TBI and add evidence to the fields of neurorehabilitation and neurorestoration.

There were no significant differences in the subgroup analysis of patients aged over 50 years old and the initial severe group between the acupuncture and non-acupuncture users after adjusting for all covariates in Table 5, which suggested that acupuncture might have less effect on consciousness recovery in older and severe TBI patients. Several studies have demonstrated that an older age and poor initial severity result in poorer disease outcomes, a higher mortality, and an inability to achieve the expected therapeutic effects, because these often accompany several complications and take longer to recover [3,33,34,35]. For groups that are more serious at the beginning, we must rely on our advanced emergency treatment guidelines. Patients can get out of the bad disease state faster and receive various treatments, including acupuncture, to achieve a better prognosis and regain consciousness. With the advent of an aging society, more and more elderly TBI patients will face the burden of treatment. Therefore, the effectiveness of acupuncture for severe TBI and older people still needs further investigation.

There were some limitations in our study. First, the generalizability of our study was limited due to our subjects being only from the North Branch of the Chang Gung Memorial Hospital in Taiwan. Second, information bias might exist due to the use of institutional registry databases that rely on accurate and comprehensive documentation of ICD-9 and ICD-10 codes. Third, we excluded subjects with incomplete first and last GCS records and GCS records that could not be quantified in terms of score, which might have led to selection bias in the results. Further rigorous clinical trials are needed to confirm the actual effectiveness of acupuncture. Fourth, the CGRD system of our hospital could not access the records of each acupuncture point. We were unable to assess the efficiencies of different acupuncture strategies.

## 5. Conclusions

Based on the rapid disease progression nature of acute TBI. It is difficult to conduct a clinical trial of acupuncture treatment and integrate acupuncture treatment into our current healthcare system. Our findings suggested that acupuncture treatment is effective for the recovery of consciousness. Conventional treatment combined with acupuncture tends to reach better outcomes compared to conventional treatment alone. Acupuncture treatment may have benefits in motor response recovery, which is related to overall consciousness recovery. Our findings also suggested that patients with TBI may choose acupuncture as an adjuvant therapy combined with conventional treatment for consciousness recovery. We hope that our findings will help to develop an integrated treatment and healthcare strategy for TBI and its complications. However, the detailed treatment mechanisms and larger-scale rigorous clinical trials still need further research and conduction.

## Figures and Tables

**Figure 1 healthcare-11-02267-f001:**
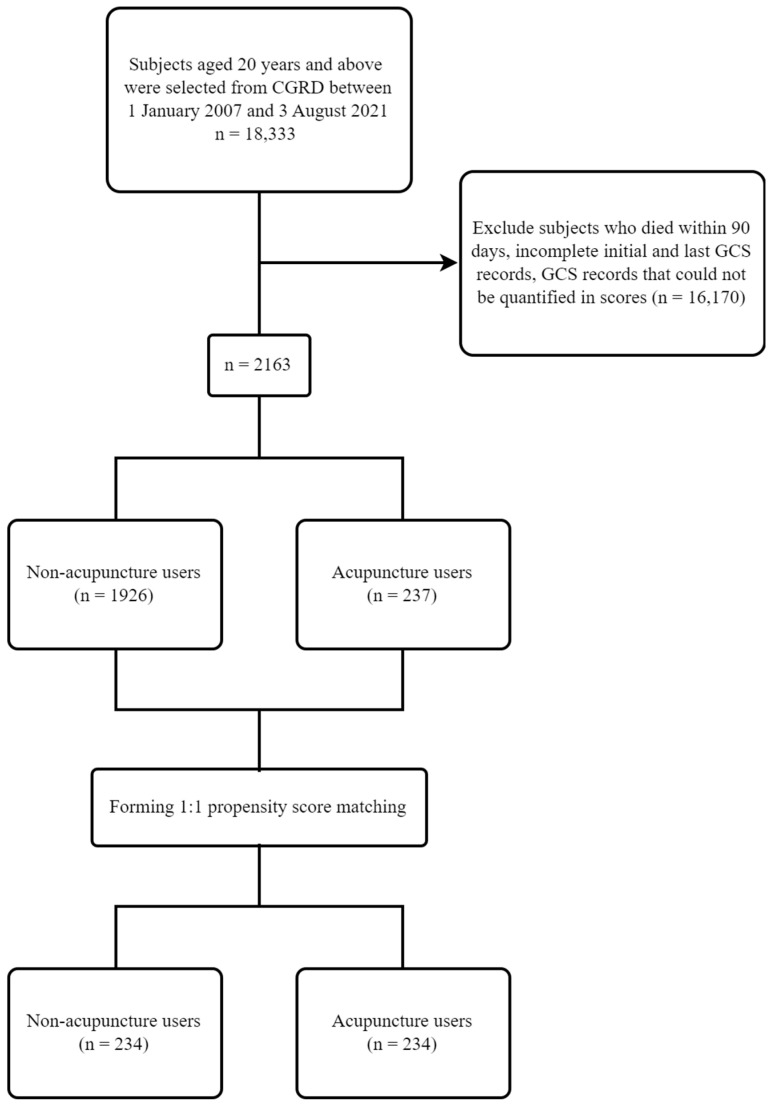
Retrospective multi-institutional cohort study flow chart.

**Table 1 healthcare-11-02267-t001:** Baseline characteristics of TBI patients between acupuncture and non-acupuncture user.

	Non-Acupuncture (*n* = 1926)	Acupuncture (*n* = 237)	*p*-Value
Gender, *n* (%)			0.076
Male	1313 (68.2)	175 (73.8)	
Female	613 (31.8)	62 (26.2)	
Age, M (IQR), year	54 (38)	57 (32)	0.417
<50	861 (44.7)	92 (38.8)	0.085
≥50	1065 (55.3)	145 (61.2)	
Initial GCS, M (IQR), score	14 (4)	11 (5)	<0.001 *
Initial GCS subgroups ^#^, *n* (%)			
Mild (13–15)	1278 (66.4)	96 (40.5)	<0.001 *
Moderate (9–12)	465 (24.1)	89 (37.6)	<0.001 *
Severe (3–8)	183 (9.5)	52 (21.9)	<0.001 *
TBI types, *n* (%)			
Skull fracture	789 (41.0)	55 (23.2)	<0.001 *
Brain contusion	92 (4.8)	16 (6.8)	0.188
SAH	520 (27.0)	72 (30.4)	0.271
SDH	724 (37.6)	110 (46.4)	0.008 *
EDH	171 (8.9)	21 (8.9)	0.993
ICH	438 (22.7)	91 (38.4)	<0.001 *
Other	68 (3.5)	8 (3.4)	0.903
Co-morbidities, *n* (%)			
Hypertension	306 (15.9)	53 (22.4)	0.011 *
Diabetes mellitus	173 (9.0)	31 (13.1)	0.042 *
Hyperlipidemia	26 (1.4)	6 (2.5)	0.153
Coronary artery disease	423 (22.0)	68 (28.7)	0.020 *
Stroke	423 (22.0)	68 (28.7)	0.020 *
Dementia	15 (0.8)	2 (0.8)	0.709
Mental disease	3 (0.2)	0 (0.0)	1.000
Parkinsonism	4 (0.2)	0 (0.0)	1.000

Note. GCS = Glasgow Coma Scale; TBI = traumatic brain injury; SAH = subarachnoid hemorrhage; SDH = subdural hemorrhage; EDH = epidural hemorrhage; ICH = intracranial hemorrhage; and M (IQR) = median (interquartile range). * *p*-value < 0.05; ^#^ Mild = 13–15; Moderate = 9–12; and Severe = 3–8.

**Table 2 healthcare-11-02267-t002:** Baseline characteristics of TBI patients after 1:1 propensity score matching between acupuncture and non-acupuncture users.

	Non-Acupuncture (*n* = 234)	Acupuncture (*n* = 234)	*p*-Value
Gender, *n* (%)			0.106
Male	156 (66.7)	172 (73.5)	
Female	78 (33.3)	62 (26.5)	
Age, M (IQR), year	55 (36)	57 (32)	0.661
<50	92 (39.3)	91 (38.9)	0.925
≥50	142 (60.7)	143 (61.1)	
Initial GCS, M (IQR), score	12 (5)	11 (5)	0.787
Initial GCS subgroups ^#^, *n* (%)			
Mild (13–15)	101 (43.2)	96 (41.0)	0.640
Moderate (9–12)	78 (33.3)	89 (38.0)	0.289
Severe (3–8)	55 (23.5)	49 (20.9)	0.505
TBI types, *n* (%)			
Skull fracture	48 (20.5)	55 (23.5)	0.435
Brain contusion	8 (3.4)	16 (6.8)	0.094
SAH	78 (33.3)	71 (30.3)	0.487
SDH	111 (47.4)	108 (46.2)	0.781
EDH	27 (11.5)	21 (9.0)	0.361
ICH	93 (39.7)	88 (37.6)	0.635
Other	9 (3.9)	8 (3.4)	0.805
Co-morbidities, *n* (%)			
Hypertension	43 (18.4)	51 (21.8)	0.356
Diabetes mellitus	27 (11.5)	29 (12.4)	0.776
Hyperlipidemia	2 (0.9)	6 (2.6)	0.285
Coronary artery disease	59 (25.2)	66 (28.2)	0.465
Stroke	59 (25.2)	66 (28.2)	0.465
Dementia	2 (0.9)	2 (0.9)	1.000
Mental disease	1 (0.4)	0 (0.0)	1.000
Parkinsonism	1 (0.4)	0 (0.0)	1.000

^#^ Mild = 13–15; Moderate = 9–12; and Severe = 3–8.

**Table 3 healthcare-11-02267-t003:** The difference in subjects’ distribution between non-acupuncture and acupuncture based on the subgroups of “Significant improvement”, “Stable condition”, and “Disease progression”.

	All Subjects		*p*-Value	1:1 PSM		*p*-Value
	Non-Acupuncture (*n* = 1926)	Acupuncture (*n* = 237)		Non-Acupuncture (*n* = 234)	Acupuncture (*n* = 234)	
Significant improvement, *n* (%)	432 (22.4)	109 (46.0)	<0.001 *	77 (32.9)	106 (45.3)	0.020 *
Stable condition, *n* (%)	1325 (68.8)	104 (43.9)		131 (56.0)	104 (44.4)	
Disease progression, *n* (%)	169 (8.8)	24 (10.1)		26 (11.1)	24 (10.3)	

* *p*-value < 0.05, significant improvement = GCS difference scores ≥ 3, stable condition = GCS difference scores between 0 and 2, and disease progression = GCS difference scores < 0.

**Table 4 healthcare-11-02267-t004:** The chance of GCS significant improvement between non-acupuncture and acupuncture groups.

	All Subjects				1:1 PSM			
	OR (95% CI)	*p*-Value	aOR (95% CI)	*p*-Value	OR (95% CI)	*p*-Value	aOR (95% CI)	*p*-Value
Non-acupuncture	1 (reference)		1 (reference)		1 (reference)		1 (reference)	
Acupuncture	2.95 (2.23–3.86)	< 0.001 *	2.01 (1.39–2.90) ^#^	< 0.001 *	1.69 (1.16–2.46)	0.006	2.11 (1.31–3.40) ^†^	0.002 *

* *p*-value < 0.05. ^#^ Adjusted for all covariates in Table 1. ^†^ Adjusted for all covariates in Table 2.

**Table 5 healthcare-11-02267-t005:** The chance of GCS significant improvement of all subjects based on gender, age, and initial severity between the non-acupuncture and acupuncture groups.

	OR (95% CI)	*p*-Value	aOR (95% CI)	*p*-Value
Gender ^†^				
Male	2.43 (1.75–3.36)	<0.001 *	1.72 (1.12–2.64)	0.013 *
Female	4.89 (2.85–8.37)	<0.001 *	3.67 (1.68–8.04)	0.001 *
Age, years ^‡^				
<50	4.85 (3.11–7.56)	<0.001 *	3.76 (2.01–7.05)	<0.001 *
≥50	2.11 (1.47–3.03)	<0.001 *	1.38 (0.87–2.18)	0.171
Initial GCS subgroups ^§^				
Severe (3–10)	1.56 (0.92–2.63)	0.009 *	1.63 (0.93–2.87)	0.088
Mild (11–15)	2.59 (1.64–4.09)	<0.001 *	2.05 (1.27–3.32)	0.004 *

* *p*-value < 0.05. ^†^ Adjusted for all covariates in Table 1 except sex. ^‡^ Adjusted for all covariates in Table 1 except age. ^§^ Adjusted for all covariates in Table 1 except the initial GCS subgroup.

**Table 6 healthcare-11-02267-t006:** The difference in subjects’ distribution between acupuncture and non-acupuncture in the category of motor response based on the subgroups of “Significant improvement”, “Stable condition”, and “Disease progression”.

	All Subjects		*p*-Value	1:1 PSM		*p*-Value
	Non-Acupuncture (*n* = 1926)	Acupuncture (*n* = 237)		Non-Acupuncture (*n* = 234)	Acupuncture (*n* = 234)	
Significant improvement, *n* (%)	54 (2.8%)	30 (12.7)	<0.001 *	7 (3.0)	29 (12.4)	<0.001 *
Stable condition, *n* (%)	1731 (89.9)	185 (78.1)		196 (83.8)	183 (78.2)	
Disease progression, *n* (%)	141 (7.3)	22 (9.3)		31 (13.2)	22 (9.4)	

* *p*-value < 0.05. Significant improvement = GCS motor response difference scores ≥ 2. Stable condition = GCS motor response difference scores between 0 and 1. Disease progression = GCS motor response difference scores < 0.

## Data Availability

Data are available in a public, open-access repository. All data relevant to the study are included in the article.

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
