# Peer review of "Effect of Acupuncture on Recovery of Consciousness in Patients with Acute Traumatic Brain Injury: A Multi-Institutional Cohort Study"

_healthcare, 2023, doi:10.3390/healthcare11162267_

Round 1
Reviewer 1 Report
Comments:
It should be discussed, that the GCS might be too insensitive to detect changes in the level of consciousness. Furthermore eye opening does not necessarily reflect improvement in consciouness, e.g.patients in persistent vegetative state may show eye opening despite haveing no awareness.
Especially in the mild TBI group (GCS 10-15) estimation of consciousness with the GCS is rather dependent on the examiner.
Also the retrospective nature of the study implies a high incertainty for correct documentation of the GCS.
What was the indication in some patients to perform acupuncture?
line 78 The study flow chart is instead of was demonstrated
line 79: Patients with newly diagnosed TBI who instead of Newly diagnosed TBI who
line 81 at least one instead of at least once
Figure 1 propensity score matching instead of propensity scorematching
Line 135: Numerical data are presentend instead of were presented
Line 146: data are expressed instead of were expressed
Line 158: ever?
Line 158 - 165: Sentence too long, verbs do not fit
Line 183: both in all subjects ....and in 1:1 PSM subjects instead of whether and or
Line 180: Table 5 indicates instead of indicated
Line 217-218: The sentence with according seems to be incomplete, verb missing
Line 217-221 cannot be understood
Line 233: Our experimental results.. To which experiments is referred to?
Reviewer 2 Report
The aim of this study was to examine the effectiveness of acupuncture treatment on recovery of consciousness in the acute phase in patients who have experienced a TBI.
Line 21: "...237 ever received acupuncture." Does this mean they had received acupuncture during this treatment period or at some time prior? Please specify.
At the start of the Introduction it would be helpful to include an explanation of the levels of TBI from mild to severe.
Lines 51-52: It is not reasonable to say that acupuncture is commonly accepted by patients as this suggests widespread acceptance. You can say that its use has become more common in recent decades.
Lines 45-47: This statement is specific to more severe levels of injury. Please be specific in referencing the review article by Galgano et al.
Line 57: Change "...in the meta-analysis." to "...in meta-analyses."
Lines 63-64: Change "...when consciousness disturbance happens." to " "..when a disturbance in consciousness is evident."
Line 68: Change "patient's" to "patients'"
Lines 105-106: Those who received less than 6 acupuncture treatments should be screened out of the study rather than classified as "non-acupuncture users."
Line 113: What is the justification for the age grouping?
The tables are comprehensive. This should be reflected in the text. The levels of TBI are confusing in the text but separated in Tables 1 & 2. Please update the text to explain the data more clearly.
Lines 208-209: The first sentence of the Discussion suggests that you are looking at acupuncture soon after TBI, not before.
Lines 210-121: It is good to see the authors have not overstated the findings.
Lines 311-312: The therapeutic benefit for acupuncture in this study is based on correlational analyses. The authors should be careful not to overstate the findings. There may be some benefit but, this has not been shown in any convincing way. so, the authors should use "may have benefit" rather than "treatment is effective"
The Discussion is generally well written.
Just a few minor suggestions regarding English language. Please see previous comments.
